# The Impact Mechanism of Household Financial Debt on Physical Health in China

**DOI:** 10.3390/ijerph20054643

**Published:** 2023-03-06

**Authors:** Jiru Song, Mingzheng Hu, Shaojie Li, Xin Ye

**Affiliations:** 1School of Statistics, Shandong University of Finance and Economics, Jinan 250002, China; 2China Center for Health Development Studies, Peking University, Beijing 100191, China; 3Institute for Global Public Policy, LSE-Fudan Research Centre for Global Public Policy, Fudan University, Shanghai 200433, China

**Keywords:** household financial debt, physical health, China, developing countries, mechanism

## Abstract

In recent years, Chinese household financial debt has been growing rapidly due to the expansion of mortgage lending. This study aims to examine the impact mechanism of Chinese household financial debt on physical health. Using the 2010–2018 China Household Tracking Survey (CFPS) panel data, we developed fixed effects models to explore the effect of household financial debt on individuals’ psychical health, and we also used an instrumental variable to address endogeneity. The findings suggest that there is a negative effect of household financial debt on physical health and these findings still hold after a series of robustness tests. In addition, household financial debt can affect individuals’ physical health through mediating variables, such as healthcare behaviors and mental health, and the effects are more significant for those who are middle-aged, married, and with low-income levels. The findings of this paper are important for developing countries to clarify the relationship between household financial debt and population health, and to develop appropriate health intervention policies for highly indebted households.

## 1. Introduction

Health is important for human beings, and it is also an important means to promote sustainable socio-economic development [1]. In the context of social issues such as the COVID-19 pandemic and aging populations, governments are incorporating health improvements into sustainable development goals [2]. China, with one-fifth of the world’s population, has the largest elderly population in the world [3]. The increasing morbidity and an aging population place a considerable burden on health and social care systems [4]. Thus, the Chinese government attaches great importance to population health, triggering the 14th Five-Year Plan for National Health in May 2022, which plans to build a better healthcare system and improve population health by 2025 [5].

Health can be affected by the economic situation. The relationship between health and the economic situation can be analyzed from the macro regional perspective and the micro individual perspective [6]. At the macro level, the rapid economic development of a region may lead to more government financial health expenditures, and may also lead to environmental pollution and ecological damage, thus affecting the health of individuals [7]. At the micro level, the deterioration of a personal economic situation may mean an increase in work pressure, a decrease in social and economic status, and a decrease in health investment, thus affecting the health of individuals [8,9].

Household financial debt is an important indicator of the economic situation [10]. Household financial debt refers to the borrowing funds of families, including all debts, bank loans, bills payable, etc. owed by all family members [11]. In China, under the influence of traditional culture, Chinese people usually borrow money to buy houses [9], so a large number of Chinese families have more or less family housing liabilities. At the same time, China’s highly accommodative monetary policy in recent years has contributed to the high leverage of Chinese households. In 2020, Chinese household financial debt reached 33.19 trillion yuan (about $4.8 trillion), 13.5 times that of 2008 [12]. High household financial debt may have a significant impact on the individuals’ social life [13]. It affects household consumption [14], the household education expenditure [15], public expenditure [16], employment [17], income inequality [18], etc. However, it is often overlooked in studies on the factors affecting the health of individuals.

Among the current research on the relationship between debt and health [8,19,20,21,22], researchers have analyzed the correlation between debt and mental health, mainly using samples from developed countries. For example, Sweet et al. (2013) found that people with high financial debt were associated with more depressive symptoms in the United States [8]. Keese (2011) used German data and found that household financial debt indicators were related to mental health and obesity [19]. Lenton et al. (2008) conducted a study on UK individuals and found a correlation between household financial debt and mental health [20]. Margareta (2018) found that household financial debt and payment difficulties were detrimental to people’s mental health in Sweden [21]. Karen (2014) found that debt had a negative impact on depressive symptoms and mental health of middle-aged and elderly people in the United States [22].

These studies from different countries have made many contributions to clarifying the impact of household financial debt on health, but they may have the following limitations: First, although existing studies have analyzed the impact of household financial debt on health, few have explored the intermediary mechanism and population heterogeneity. However, sorting out the intermediary mechanism of the impact of household financial debt on health and analyzing whether this correlation is different among different populations can help us understand the impacts more clearly. Second, existing research mainly carries out correlation analysis, but few studies consider the endogenous problems such as missing variables and reverse causality, and it is difficult to conclude causal inference. Third, existing research has mainly explored the effects of family debt on mental health, while neglecting the effects on physical health. However, physical health is different from mental health [23]. Physical health is defined as the absence of disease in the organs, a high resistance to disease, and a high capacity for physical activity and work [24]. In contrast, mental health is defined as the absence of mental illness [25], and living in a state of well-being [26]. Fourthly, most of the available studies have taken developed countries as their samples. However, there may be significant differences in the demand for loans, household income, and consumption levels in developing countries, so the findings of existing studies may not be applicable to developing countries such as China [27,28].

In this paper, we explore the impact mechanism of household financial debt on people’s physical health in China, the world’s largest developing country. Instrumental variables are used to address endogeneity. In addition, the mediating roles of healthcare behaviors and mental health were further analyzed, and heterogeneity was analyzed for people with different ages, marital statuses, and incomes. The findings of this paper are important for developing countries to clarify the relationship between household financial debt and population health, and then to formulate appropriate health intervention policies for different indebted households.

## 2. Theoretical Analysis and Hypothesis Development

Dahlgren and Whitehead put forward the theory of social determinants of health in 1991 [29]. This theory reckons that in addition to factors that directly cause disease, people’s social status, as well as living and working environment, such as people’s income, wealth, economic status, living conditions, etc., are also factors affecting health. According to this theory, household financial debt, as a reflection of people’s wealth and economic status, should also be one of the social determinants of health. Therefore, this paper proposes research hypothesis 1.

**H1.** *Household financial debt has a negative impact on physical health*.

Additionally, in terms of the mediating mechanism of the effect of household financial debt on physical health, this paper argues that household financial debt affects individuals’ physical health in two main ways. (1) Healthcare behaviors: when people have household financial debt, they will spend less on consumption [30], and therefore they will spend less on health care. This means that they may choose to seek health care at cheaper, lower-quality facilities, and then diseases may not be effectively treated, which can be detrimental to physical health. (2) Mental health: previous studies have shown that family debt has a negative impact on mental health [8,19,20,21,22], while additional epidemiological studies have shown that mental health indicators, such as depression, stress, and poor sleep quality, are significantly associated with physical health indicators, such as obesity and hypertension [23]. Therefore, this paper further proposes the following two research hypotheses.

**H2.** *Healthcare behaviors mediate the effect of household financial debt on physical health*.

**H3.** *Mental health mediates the effect of household financial debt on physical health*.

Moreover, the impact of household financial debt on physical health may be heterogeneous across populations. First, concerning different age groups in China, the middle-aged population is the main workforce of the family and bears most of the responsibility for repaying household financial debts, whereas young people and older people are in the cared-for group and often do not have to bear the responsibility for repaying debts [31]. Therefore, the impact of household financial debt on the physical health of middle-aged people may be greater than that of young and old people. Secondly, influenced by traditional culture, Chinese people have the habit of using family debt to buy a house when they get married [9]. Therefore, married people tend to have more pressure to repay household financial debt, and the impact of household financial debt on the physical health of married people may be greater. Finally, income is an important source of funds for debt repayment [32]. People at low-income levels may face greater pressure to repay debt, and the impact of household financial debt on physical health may be greater for people at low-income levels. Therefore, this paper proposes the following three research hypotheses.

**H4.** *Household financial debt has a greater impact on physical health in middle-aged adults compared to young and old adults*.

**H5.** *Household financial debt has a greater impact on physical health among married people compared to unmarried people*.

**H6.** *Household financial debt has a greater impact on the physical health of people with lower income levels compared to those with higher income levels*.

## 3. Materials and Methods

### 3.1. Data Source

The data used in this paper are from the China Family Panel Studies (CFPS). CFPS is a nationwide survey organized and implemented by the China Social Science Survey Center of Peking University, which reflects China’s social and economic changes by tracking data at the individual, household, and village levels, covering a wide range of information on the economy, education, and health [33]. CFPS has been approved by the Biomedical Ethics Review Committee of Peking University (ID: IRB00001052-14010). CFPS has been conducted every two years since the first official survey in 2010, and now has five years of complete survey data. For the 2010 CFPS, a multistage probability distribution was used to stratify the sample. As a result, five provinces/regions (Gansu, Guangdong, Henan, Liaoning, and Shanghai) were selected for initial over-sampling (1600 households in each, or an aggregate of 8000) to obtain regional comparisons, and another 8000 households were selected by weighting from the remaining provinces/regions to make the overall CFPS sample nationally representative [34].

This paper used the longitudinal data from CFPS of 2010, 2012, 2014, 2016, and 2018. We merged the adult database with the family economic database. We excluded observations that answered “unable to judge”, “missing”, “not applicable”, “refused to answer”, “don’t know”, or “no data” for the independent, dependent, and mediating variables used in our study.

### 3.2. Measures

#### 3.2.1. Explained Variable: Physical Health

Following previous literature, this survey used a self-rated health status to measure physical health [35]. The CFPS data includes a question on the subjective physical health status of individuals, asking “How do you think your health is?” and the data are tabulated by assigning a value to the self-rated health in ranked order, from 1 to 5. The higher the score, the better the self-rated health of the population.

In addition, we chose to test for robustness by using proxy variables. We chose the following variables as a proxy for the physical health variables:

(1) Whether individuals have been physically unwell in the past two weeks

The CFPS surveyed the physical health status of people with the question “Have you been physically unwell in the past two weeks?” This variable is a binary variable and is assigned a value of 1 if the answer is “yes”, otherwise it is assigned a value of 0.

(2) Whether individuals have suffered from a chronic disease in the past 6 months

The CFPS surveyed the question “Have you suffered from a chronic disease diagnosed by a doctor in the past six months?”. These chronic diseases include diabetes, hypertension, respiratory diseases, etc. The variable is a binary variable that is assigned a value of 1 if the answer is “yes” and 0 otherwise.

(3) Whether the individual has been hospitalized in the past year

The CFPS surveyed whether individuals had been hospitalized in the past year by asking the question “Have you been hospitalized in the past year?”. This variable is a binary variable and is assigned a value of 1 if the answer is “yes”, otherwise it is assigned a value of 0.

#### 3.2.2. Core Explanatory Variables: Household Financial Debt

In this paper, household financial debt is used as the core explanatory variable. The variable takes the value of 1 if a household has financial debt and 0 if a household has no financial debt.

#### 3.2.3. Instrumental Variable: Interest Rate

The core variable of our study, household financial debt, may have an endogeneity problem because people with physical diseases may incur household financial debt due to excessive medical expenditures, and thus there may be an inverse causality between household financial debt and physical health. To address the endogeneity problem, we chose the instrumental variable method. The instrumental variable is the five-year annualized interest rate of China obtained through the OECD database. This instrumental variable was chosen for the following reasons: the economics theory on the equilibrium of the borrowing market suggests that the decline in the amount of household borrowing is a rational reflection of people’s response to lower borrowing rates [36], and therefore, there is a negative correlation between interest rates and household financial debt. At the same time, interest rates, also as the macro indicator, are set by the central government and can hardly be influenced by the individual health status [37], thus it can be considered an effective instrument variable.

#### 3.2.4. Mediating Variables

Healthcare behaviors and mental health were chosen as mediating variables.

(1) Healthcare Behaviors

The CFPS asks respondents to rate the level of healthcare at the medical site where they seek for medical care on a daily basis, on a scale of 1–5. We constructed the variable of healthcare behaviors based on this question. A higher score means the higher level of care the respondent is able to receive from their chosen medical site, and the better healthcare behaviors.

(2) Mental health

Depression is a commonly used variable to represent people’s mental health [38,39,40]. The CFPS survey includes questions such as “How often do you feel hopeless about the future?” and “How often do you find it difficult to do anything?”. The higher the score, the higher the depression level. The sum of the scores of the 8-question depression scale in the CFPS database is used to measure mental health status. The higher the sum of the scores, the more serious the depression and the worse the mental health status. We examined the reliability coefficient of the CES-D8 scale and the results showed that Cronbach’s alpha was greater than 0.7.

#### 3.2.5. Control Variables

Referring to the empirical studies on household financial debt and health status [19,41,42], we controlled variables such as household registration, gender, age, education, marital status, family size, and net income per capita. For the household registration variable, “agricultural” was assigned a value of 1 while “non-agricultural” was assigned 0. For education, “illiterate/semi-literate” was assigned a value of 0, while “elementary school” was 1, “middle school” was 2, “high school” was 3, and “university or above” was 4. For marital status, “married” was assigned the value of 1 while “unmarried” was 0. For gender, “female” was assigned as 0 while “male” was 1. For the family size variable, it refers to the number of members in a household. Moreover, for income, we took the logarithm of the net income per capita variable in CFPS, so as to reduce the impact on the regression results due to too large fluctuations in income.

### 3.3. Statistical Analysis

The statistical analysis was divided into five stages. In the first stage, we conducted a descriptive analysis. In the second stage, we explored the relationship between household financial debt and mental health using a fixed effects linear regression model with the time-fixed effects, as well as individual fixed effects [43,44,45]. In the third stage, to verify the robustness of the results, 3 approaches were chosen for robustness testing. (1) Ordered logistic regression: since our explanatory variables were ordered categorical variables, we tested the robustness of the results using ordered logistic regression, fixed effects ordered logistic regression. (2) Instrumental variables: to address the endogeneity problems, the instrumental variable was used for the 2SLS (two-stage least square) model. At the same time, to avoid the influence of heteroskedasticity, the 2SLS model was estimated using the GMM (generalized method of moments) method. (3) Replacement variables: physical health was measured by replacing “self-rated health” with “whether have been physically unwell in the past two weeks”, “whether have been chronically ill in the past six months”, and “whether have been hospitalized in the past year”. In the fourth stage, we explored the mediating effect of mental health status and health behaviors in the relationship between household financial debt and physical health. We drew the pathway map of household financial debt affecting physical health by SEMs (structural equation models), and we also tested for mediating effects using the bootstrap method. In the fifth stage, we conducted a heterogeneity analysis to explore differences in the relationship between household financial debt and physical health among people with different ages, marital statuses, and income levels, respectively.

The *p*-values below 0.05 were considered to be statistically significant. The analysis was performed using STATA statistical software package, version 16 SE.

## 4. Results

### 4.1. Descriptive Analysis

The descriptive statistics are shown in Table 1. The mean value of the explanatory variable “self-rated health” was 3.209, with a standard deviation of 1.290; the mean value of the core explanatory variable “household financial debt” was 0.350, with a standard deviation of 0.477, indicating a large variation in financial debt status between households. Additionally, 49.9% of the sample is men; 72.7% of the sample is rural households; and 77.4% of the sample is married.

### 4.2. Basic Regression Results

The fixed effects model was chosen to estimate the relationship between household financial debt and people’s physical health, and the results are shown in Table 2. As shown in the first column, the regression coefficient of household financial debt without the inclusion of control variables is −0.062 and is significant at the 1% level, indicating that people in households with debt have worse physical health status. The second column shows the regression results with the inclusion of control variables. The regression coefficient of household financial liabilities becomes −0.054 and is still significant at the 1% level. Therefore, the research hypothesis H1 was proved to be correct.

### 4.3. Robustness Tests

#### 4.3.1. Ordered Logistic Regression

Table 3 shows the regression results of the ordered logistic regression. The first column demonstrates the results without fixed effects, where the regression coefficient of household financial debt on physical health is −0.070 and is significant at the 1% level. In addition, the second column demonstrates the results with fixed effects. The regression coefficient is −0.115 and the significance of household financial debt is unchanged, which proves the robustness of the baseline regression results.

#### 4.3.2. Instrumental Variable

Table 4 shows the results of the 2SLS model regression using the GMM method. In the first stage of regression, the regression coefficient of the instrumental variable “interest rate” is −2.551 and is significant at the 1% level. The sign of the regression coefficient is consistent with the economic theory of equilibrium in the lending market [36]. Additionally, the F-statistic is 290.62, which proves the fitness of the instrumental variable. Moreover, the results of the second stage show that the coefficient of household financial debt on physical health is −0.998 and is significant at the confidence level of 1%, indicating that people in households with debt will have poorer physical health status.

#### 4.3.3. Replacement of Explanatory Variables

We replaced “self-rated health” with “whether physically unwell in the past two weeks”, “whether had chronic diseases in the past 6 months”, and “whether hospitalized in the past year”, and the regression results are shown in columns (1), (2) and (3) of Table 5 respectively. After replacing the explanatory variables, the regression results are still significant at the 1% level, indicating that the higher the level of household financial debt, the worse the physical health status of individuals, which shows the robustness of the baseline regression result again.

### 4.4. Intermediary Mechanism

#### 4.4.1. Structural Equation Models

We used structural equation models to construct a pathway diagram of the effect of household financial debt on physical health. The results are shown in Figure 1. First, the regression coefficients of household financial debt on healthcare behaviors, physical health, and mental health are significant at the 1% level, with coefficients of 0.096, −0.696, and −0.086, respectively. Second, the regression coefficient of healthcare behaviors on physical health is −0.055 and the regression coefficient is significant at the 1% level. Third, the regression coefficient of mental health on physical health is 0.298 and is significant at the 1% level.

#### 4.4.2. Bootstrap Intermediary Mechanism Test

We further chose the bootstrap mediation mechanism test to explore the mediating effect size and significance. The results are shown in Table 6. For healthcare behaviors, the indirect effect is −0.003 and the direct effect is −0.131, and both coefficients are significant at the 1% level. As for mental health, the indirect effect is −0.062 and the direct effect is −0.061, and both effects are significant at the 1% level. Therefore, the research hypotheses H2 and H2 were proved to be correct.

### 4.5. Heterogeneity Analysis

#### 4.5.1. Age

Table 7 shows the heterogeneous characteristics of the effect of household financial debt on the physical health under different age groups. Among those under 30 years of age, the regression coefficient of household financial debt on physical health is −0.028 and is significant at the 5% level. Among those aged 30–75 years, the regression coefficient of household financial debt on physical health is −0.061 and is significant at the 1% level, which suggests that household financial debt has a greater impact on the health status of middle-aged people. Moreover, the regression coefficient of household financial debt on physical health is −0.049 among those aged 75 years and older and the result was not significant. Therefore, research hypothesis H4 was proved to be correct.

#### 4.5.2. Marital Status

Table 8 shows the heterogeneous effects of household financial debt on the health status of people across marital status. For those married, the coefficient value is −0.060 and significant at the 1% level, while for those unmarried, the coefficient value is −0.018, with a weaker effect and insignificant results. Therefore, research hypothesis H5 was proved to be correct.

#### 4.5.3. Income Level

Table 9 shows the heterogeneity of the effect of household financial debt on the physical health for individuals with different income levels. For those whose income lies in the lowest 25% range, the coefficient value is −0.166 and significant at the 1% level. For those whose income levels lie between 25–50%, the coefficient value is −0.113 and significant at the 5% level. Moreover, for those whose income levels lie between 50–75% and greater than 75%, the coefficient is insignificant, and the values are −0.062 and −0.065, respectively, which indicate that the effect of household financial debt on physical health status decreases with the increase of income level of the population. Therefore, research hypothesis H6 was proved to be correct.

## 5. Discussion

Using the longitudinal data of CFPS during 2010–2018, we empirically analyzed the impact of household financial debt on individuals’ physical health, as well as explored the mediating mechanisms and the heterogeneity of different groups. The results are articulated as follows.

First, household financial debt has a significant negative effect on individuals’ physical health. That is, those in households with debt have worse physical health status. The finding holds after replacing the explanatory variables and adding an instrumental variable, which is consistent with results from developed countries, such as the United States, Germany, and the United Kingdom [8,19].

Second, household financial debt can have a negative impact on individuals’ health by affecting their healthcare behaviors and mental health. People with debt cannot afford high medical expenses and may choose cheap clinics with less satisfactory medical treatment. Therefore, diseases may not be effectively treated, which will affect physical health. This conclusion is consistent with previous research on debt and consumer spending [30]. In addition, household financial debt makes individuals bear the pressure of repayment and brings them mental health problems, such as depression. Depression and psychological burdens are often considered to be related to physical health diseases, such as hypertension and obesity. This conclusion is consistent with the research on the impact of debt on mental health [46], and the relationship between mental health and physical health [23].

Thirdly, the effects of household financial debt on physical health are heterogeneous across populations. First, the effects are more significant in the middle-aged population between the ages of 30 and 75. This may be due to the fact that middle-aged people are the main breadwinners of the family and bear the majority of household financial debts, which in turn causes severe physical discomfort and contributes to the progressive deterioration of physical health. This is in line with the findings of existing studies on the relationship between household financial debt and health [22,47]. Second, the effects are more significant for those who are married compared to those who are unmarried. This may be due to the fact that marriage pulls consumption and demand, which in turn increases the likelihood of borrowing and changes the level of household financial debt. Thus, those who are married tend to have more household financial debt and incur higher levels of psychological stress during debt repayment, which in turn causes more severe deterioration in physical health. This is consistent with the findings of existing research on debt and life satisfaction [48]. Finally, for the income level, the regression results are most significant for those with a low net household income and gradually weaken with the increase of the income level. This may be due to the fact that low-income households are faced with increasing consumption levels and constantly optimizing the quality of services and have to resort to borrowing to maintain their daily needs, which leads to a vicious cycle of deteriorating physical and mental health, resulting in the long-term suffering of both health and debt [20,49].

There are certain research limitations in this paper. First, the measurement of physical health is self-rated, which may incur self-report bias. Second, since chronically ill might also refer to chronic mental diseases and mentally ill people can also be hospitalized, these two indicators may not measure physical health accurately. However, the bias may be minimal, because few individuals in China consider chronic mental diseases as having chronic diseases, and even fewer are hospitalized for the reason of mental illness [50]. Third, more exact explanatory variables measure can be developed to reflect other perspectives of debts, such as debt structure and attitudes toward debt.

Meanwhile, the strengths of this paper are mainly reflected in the following aspects. First, while previous studies have paid less attention to theoretical mechanisms and subgroup analysis [8,20], this paper further expands to examine the mediating mechanisms, as well as population heterogeneity, and further enriches the theoretical mechanisms of action between household financial debt and individuals’ physical health. Second, existing studies have focused on the impact of family debt on mental health [8,47,51], while neglecting the impact on physical health, and this study focuses on this research gap. Third, while most previous studies have only conducted correlation studies, this paper addresses possible endogeneity issues using an instrumental variable and conducts causal inference. Fourth, most previous studies have taken developed countries as samples [8,19,20], while this paper takes China, the world’s largest developing country, as a sample, so as to provide relevant policy recommendations for other developing countries.

## 6. Conclusions

This paper investigates the impact mechanism of household financial debt on the physical health of the population and shows that household financial debt has a significant negative effect on physical health, and this finding holds after robustness tests, such as regression on instrumental variables and replacement of explanatory variables. In addition, healthcare behaviors and mental health play a mediating role, and the effects are heterogeneous across households, age, marital status, and income levels. The policy inspiration of this paper is that the government and community should focus on the health status of highly indebted households, especially for the middle-aged, married, and low-income groups, and provide health education to highly indebted individuals and hold regular communication sessions to guide individuals to communicate their problems and improve their health.

## Figures and Tables

**Figure 1 ijerph-20-04643-f001:**
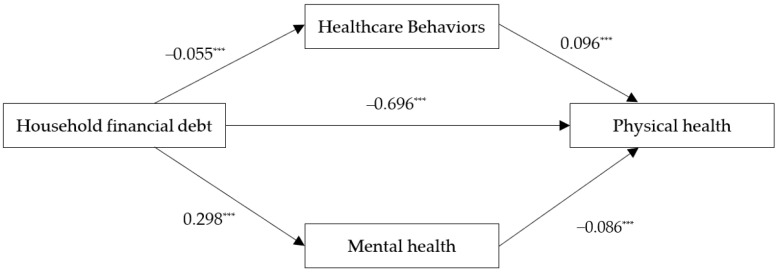
Pathway diagram of the effect of household financial debt on physical health. Note: *** *p* < 0.01.

**Table 1 ijerph-20-04643-t001:** Descriptive statistics.

Variables	N	Mean/%	SD	Min	Max
Self-rated health	176,534	3.209	1.290	1	5
Household financial debt	176,977	0.350	0.477	0	1
Interest rate	176,977	5.80%	0.008	4.90%	6.78%
Mental health	96,762	13.998	3.876	8	32
Healthcare behaviors	134,532	3.408	0.772	1	5
Gender	178,895	49.9%	0.500	0	1
Age	176,963	45.014	17.558	9	110
Household registration	165,515	72.7%	0.445	0	1
Marital status	170,514	77.4%	0.418	0	1
Education	174,759	1.575	1.261	0	4
Ln(Net income per capita)	171,403	9.164	1.146	−1.609	15.549
Family size	176,977	4.333	1.968	1	26

**Table 2 ijerph-20-04643-t002:** Regression results of household financial debt on physical health.

	Physical Health	
	(1)	(2)
	Fixed Effect	Fixed Effect
Household financial debt	−0.062 ***	−0.054 ***
	(−24.45)	(0.007)
Constant term	4.363 ***	4.030 ***
	(312.58)	(0.334)
Control variables		√
Year fixed effects	√	√
Individual fixed effects	√	√
R^2^	0.1332	0.1557
N	176,534	158,671

Note: *** *p* < 0.01. Control variables include household registration, gender, age, education, marital status, family size, and net income per capita. √ means that these variables or effects are controlled.

**Table 3 ijerph-20-04643-t003:** Ordered logistic regression results.

	Physical Health	
	(1)	(2)
	Ologit	FE-Ologit
Household financial debt	−0.070 ***	−0.115 ***
	(0.015)	(0.017)
Control variables	√	√
Year fixed effects		√
Individual fixed effects		√
R^2^	0.042	0.320
N	158,671	136,008

Note: *** *p* < 0.01. Control variables include household registration, gender, age, education, marital status, family size, and net income per capita. √ means that these variables or effects are controlled.

**Table 4 ijerph-20-04643-t004:** Regression results for instrumental variable.

	Physical Health
Household financial debt	−0.547 ***
	(0.137)
Constant term	4.819 ***
	(0.051)
R^2^	0.108
First-stage results	
Interest rate	−2.551 ***
	(0.150)
Constant term	0.499 ***
	(0.019)
First stage SW-F statistic	290.62
Control variables	√
N	158,671

Note: *** *p* < 0.01. Control variables include household registration, gender, age, education, marital status, family size, and net income per capita. √ means that these variables are controlled.

**Table 5 ijerph-20-04643-t005:** Regression results after replacing the explanatory variables.

	(1)	(2)	(3)
	Physical Unwell	Chronic Diseases	Hospitalized
Household financial debt	0.033 ***	0.011 ***	0.011 ***
	(0.003)	(0.002)	(0.002)
Constant term	−0.080	0.082	0.260 ***
	(0.140)	(0.095)	(0.073)
Control variables	√	√	√
Year fixed effects	√	√	√
Individual fixed effects	√	√	√
R^2^	0.045	0.034	0.007
N	151,956	151,935	155,535

Note: *** *p* < 0.01. Control variables include household registration, gender, age, education, marital status, family size, and net income per capita. √ means that these variables or effects are controlled.

**Table 6 ijerph-20-04643-t006:** Intermediary mechanism test.

Variables	Indirect Effect	Direct Effect
Healthcare behaviors	−0.003 ***	−0.131 ***
	(0.001)	(0.007)
Mental health	−0.062 ***	−0.061 ***
	(0.003)	(0.008)

Note: *** *p* < 0.01. Control variables include household registration, gender, age, education, marital status, family size, and net income per capita.

**Table 7 ijerph-20-04643-t007:** Regression results of age heterogeneity.

	Physical Health		
	(1)	(2)	(3)
Age	<30	30–75	>75
Household financial debt	−0.028 **	−0.061 ***	−0.049
	(0.013)	(0.008)	(0.057)
Constant term	5.378 ***	3.369 ***	−3.930
	(0.404)	(0.528)	(3.918)
Control variables	√	√	√
Year fixed effects	√	√	√
Individual fixed	√	√	√
R^2^	0.158	0.117	0.117
N	38,143	116,176	5989

Note: ** *p* < 0.05, *** *p* < 0.01. Control variables include household registration, gender, age, education, marital status, family size, and net income per capita. √ means that these variables or effects are controlled.

**Table 8 ijerph-20-04643-t008:** Regression results of marital heterogeneity.

	Physical Health	
	(1)	(2)
Marital Status	Unmarried	Married
Household financial debt	−0.018	−0.060 ***
	(0.015)	(0.008)
Constant term	4.590 ***	3.758 ***
	(0.408)	(0.420)
Control variables	√	√
Year fixed effects	√	√
Individual fixed	√	√
R^2^	0.280	0.124
N	34,837	123,834

Note: *** *p* < 0.01. Control variables include household registration, gender, age, education, family size, and net income per capita. √ means that these variables or effects are controlled.

**Table 9 ijerph-20-04643-t009:** Regression results of income heterogeneity.

	Physical Health			
	(1)	(2)	(3)	(4)
Income Quartiles	1	2	3	4
Household financial debt	−0.166 ***	−0.113 **	−0.062	−0.065
	(0.045)	(0.046)	(0.050)	(0.038)
Constant term	4.874 ***	1.657	5.639 **	−0.963
	(1.264)	(7.100)	(2.224)	(2.390)
Control variables	√	√	√	√
Year fixed effects	√	√	√	√
Individual fixed	√	√	√	√
R^2^	0.303	0.0002	0.235	0.0003
N	15,657	17,177	17,383	15,725

Note: ** *p* < 0.05, *** *p* < 0.01. Control variables include household registration, gender, age, education, marital status, family size, and net income per capita. √ means that these variables or effects are controlled.

## Data Availability

The CFPS datasets for this study can be found at http://www.isss.pku.edu.cn/cfps/ (accessed on 30 November 2022).

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
