# Peer review of "The Impact Mechanism of Household Financial Debt on Physical Health in China"

_ijerph, 2023, doi:10.3390/ijerph20054643_

Round 1

Reviewer 1 Report (Previous Reviewer 1)

It is a good paper, but the I think the new title should be The Impact Mechanism of Household Financial Debt on Physical Health in China not The Impact and Mechanism of Household Financial Debt on Physical Health in China, while the content refers to the mechanism of impact and not separately to impact and mechanism.

Author Response

It is a good paper, but the I think the new title should be The Impact Mechanism of Household Financial Debt on Physical Health in China not The Impact and Mechanism of Household Financial Debt on Physical Health in China, while the content refers to the mechanism of impact and not separately to impact and mechanism.

Response : Thank you so much for taking your time and effort to review our manuscript. Following your kind suggestion, we have revised the title to “The Impact Mechanism of Household Financial Debt on Physical Health in China”.

Reviewer 2 Report (Previous Reviewer 3)

Let me say that I was positively surprised on this new version. The authors I have made very strong efforts in revising their work, overall. They satisafctorily replied to all my points, and they were many and not so easy to solve out, even though I am still thinking that the 3-step analytical approach proposed by the authors is tricky. They might have applied SEM models also in longitudinal way and thus modeling both the mediations and the instrumental variables simultaneously with controlling for the sphericity of the variables across the waves of times. All in all, the complex 3-step sequence of analyses seems hold. Finally, I appreciated the work on the discussion and conclusions that makes this work more interesting than was at the beginning. Please draw the attention to some English typos into the manuscript.

Author Response

Let me say that I was positively surprised on this new version. The authors I have made very strong efforts in revising their work, overall. They satisafctorily replied to all my points, and they were many and not so easy to solve out, even though I am still thinking that the 3-step analytical approach proposed by the authors is tricky. They might have applied SEM models also in longitudinal way and thus modeling both the mediations and the instrumental variables simultaneously with controlling for the sphericity of the variables across the waves of times. All in all, the complex 3-step sequence of analyses seems hold. Finally, I appreciated the work on the discussion and conclusions that makes this work more interesting than was at the beginning. Please draw the attention to some English typos into the manuscript.

Response : Thank you so much for taking your time and effort to review our manuscript. We appreciate all your valuable comments and suggestions, which are very helpful in improving the quality of our paper. We have invited a professional English writer to revise our paper in order to avoid confusing or wrong English expressions.

Reviewer 3 Report (Previous Reviewer 4)

Well done, Authors addressed all comments correctly

Author Response

Well done, Authors addressed all comments correctly

Response : Thank you so much for taking your time and effort to review our manuscript. We appreciate all your valuable comments and suggestions, which are very helpful in improving the quality of our paper.

This manuscript is a resubmission of an earlier submission. The following is a list of the peer review reports and author responses from that submission.

Round 1

Reviewer 1 Report

The subject of the paper is an interesting one, although as the authors confirm the analysis has significant limitations. Overall it is an interesting paper, but there are also some issues to deal with, as follows:

1.      The paper should be revised to avoid confusing or wrong English expressions.

2.      The authors need to clarify what is the meaning of physical health they have considered, compared to the meaning of mental health.

3. If physical health is interpreted totally separate from mental health, as suggested in introduction, then how can we be sure that answers to questions like “whether have been unwell in the past two weeks”, “whether have been chronically ill in the past six months”, and “whether have been hospitalized in the past year” can be considered exclusively to refer to only physical health (as the authors sustain in page 4). For example, since unwell might mean also mental issues, chronically ill might refer to chronic mental diseases and mentally ill people are also hospitalized then at least some of the responses may refer to mental health instead of physical health.

Author Response

Response to Reviewer 1 Comments The subject of the paper is an interesting one, although as the authors confirm the analysis has significant limitations. Overall it is an interesting paper, but there are also some issues to deal with, as follows: Response : Thank you so much for taking your time and effort to review our manuscript. We appreciate all your valuable comments and suggestions, which are very helpful in improving the quality of our paper. All the comments raised have been addressed properly in the revised manuscript, which have been clearly marked in red in the paper. Please find our itemized responses below and revised manuscript in the re-submitted files. Point 1: The paper should be revised to avoid confusing or wrong English expressions. Response 1: Thank you for your suggestions. We have invited a professional English writer to revise our paper in order to avoid confusing or wrong English expressions. Point 2: The authors need to clarify what is the meaning of physical health they have considered, compared to the meaning of mental health. Response 2: Thanks for your suggestions. We have clarified the concepts of pyhsical health compared to mental health that “physical health is different from mental health (Ohrnberger, Fichera et al. 2017). Physical health is defined as the absence of disease in the organs, a high resistance to disease, and a high capacity for physical activity and work (Callahan 1973). In contrast, mental health is defined as the absence of mental illness (Rockville 1999), and living in a state of well-being (World Health Organization 2004).” (Line 87-91). Point 3: If physical health is interpreted totally separate from mental health, as suggested in introduction, then how can we be sure that answers to questions like “whether have been unwell in the past two weeks”, “whether have been chronically ill in the past six months”, and “whether have been hospitalized in the past year” can be considered exclusively to refer to only physical health (as the authors sustain in page 4). For example, since unwell might mean also mental issues, chronically ill might refer to chronic mental diseases and mentally ill people are also hospitalized then at least some of the responses may refer to mental health instead of physical health. Response 3: Thank you for your comment. Actually, the first question is “whether have been physically unwell in the past two weeks”. So it refers to physical health. For the second and third question “whether have been chronically ill in the past six months”, and “whether have been hospitalized in the past year”, few residents in China consider mental diseases as chronic diseases, and even fewer are hospitalized because of mentally ill (Lu, Xu et al. 2021). Therefore, it can be assumed that these three questions refer mainly to physical health. Actually, in studies using CFPS data, a lot of researchers have usually used these indicators to measure physical health (Yang, Wang et al. 2022). (Line 178-179). In addition, we have acknowledged possible confounding by mental illness in the measurement of physical health in our limitations, but the effect on the results can be minimal. (Line 452-456).

Reviewer 2 Report

This is an interesting paper with an impressive sample size which has the potential to make a significant contribution to the literature. However there are several issues which need addressing. My main concerns are about some detail lacking in the methodology, and some possible errors and more detail/clarity needed for some of the statistical analysis.

Title

·         I wonder if you might consider changing the title to incorporate the aim around mediating variables and demographic differences (although see comment below I don’t think this actually mediation at the moment).

Abstract

·         Add in background and aims sentences.

·         Line 19: The effect is more significant: Please consider re-wording to stroner.

Introduction

·         Line 26: Please consider expanding why health is important beyond economic reasons.

·         Line 31: Please add briefly some detail you say ‘realize a healthcare system’ in China but there is already one so please be specific here.

·         First paragraph: Is there any evidence here you can cite on if China differs in population health to other countries?

·         Line 34: 33.19 trillion yuan: Please also report this in another currency such as US dollars for international readership. It would also be helpful to have this broken down into average per household if possible please.

·         You say that ‘Only few available literature on the relationship between debt and health”: Please consider rewording this sentence. More importantly: I don’t think this is correct. There is quite a lot of literature on debt and health, including meta-analyses in the area. Please rewrite the 3rd paragraph of the introduction overviewing the research before: At the moment this downplays the amount of research that has already been done.

·         Line 48: “A good basis for this paper”: Please re-word.

·         Line 56: Please define endogeneity the first time you use it.

·         Line 64: Typo (population).

Materials and Methods

·         Line 68 onwards: Please provide more detail about the recruitment and methods of the China Family Panel Studies.

·         My main issue with this section is you say about 5 time points, but it is not clear whether you are analysing all of these, and if so if the data is longitudinal i.e. if it is the same participants linked across time. Your results read like it is not and it is cross sectional, but here and throughout the paper it needs to be clearer whether your study is cross-sectional or longitudinal.

·         Line 78: You say about data removed for those who answered not applicable refused to answer etc were removed but it is unclear what questions/items this is referring to.

·         Line 82: Please change ‘We use’ to ‘This survey used’.

·         Page 83: Typo ‘physica’

·         Line 93: Was chronic disease defined in the survey, examples given etc?

·         Line 116: Please define exogenous variable the first time you use this.

·         Line 120 onwards: Mental health measure of depression. Please give details here: was this a standardised measure? If not how was it developed? What was the chronbachs alpha for this sample?

·         Line 133: Please define hukou.

·         Line 138: Was there anyone not identifying as male or female, or were there only these two binary options on the survey?

·         Statistical analysis section: Please give details if the variables were normally distributed and met the criteria for normal distribution.

Results

·         Lines 160-164: You do not need to repeat all of the means, standard deviations in the text as they are in the table. Please edit.

·         For categorical variables of gender (and possibly education) it does not make sense to report these as a mean when they are categorical, please report % instead, and consider if adding these as continuous variables into the regression models is the correct way to analyse these categorical variables.

·         For all tables: Please make clear what ‘_cons’ is.

·         For all tables: Please add in details of what control variables were added and the effect of these.

·         For all tables: I don’t think p<0.1 should be used, keep significance to p<.05 or less given the amount of tests used.

·         You say in methods about interest rates being included but this is not clear in the results where this fits (presumably one of the control variables but more detail is needed here).

·         Table 5: This is described here and in discussion as an intermediary/mediating test. However I don’t think that it is. Here you show the link between household debt and mental health and health behaviours. This is interesting and important, but it doesn’t show, as the rest of the paper suggests, a mediating role with physical health. This could be done in another way for example through statistical mediation. Please change the results to show that health behaviours and mental health do actually have an intermediary effect on physical health, or revise the rest of the paper accordingly to show that this is a separate finding.

·         For the separate analyses by age, income level and gender: These are important and interesting. However before showing that there is a different relationship between debt and physical health depending on these variables, I think you also need to show if these variables impact debt and physical health separately. For example physical health will likely vary depending on age. i.e. show that, for example, gender impacts both debt and physical health, and then that it impacts the relationship between debt and physical health.

·         Table 7 typo (psysical).

Discussion

·         Please revise discussion according to changes to results suggested above.

·         As above, please change wording around talk of mediation.

·         Please be careful with language about causality e.g. line 257 (household debt has a significant negative effect on residents’. This relates to point about being clear your data is cross-sectional. Line 267-270 is another example of where the conclusions go beyond the results, as per comments above you haven’t shown that cigarettes mediate the link between mental and physical health.

·         Add cross-sectional to limitations.

·         Conclusions: Again be cautious about using term mediating.

Author Response

Response to Reviewer 2 Comments

This is an interesting paper with an impressive sample size which has the potential to make a significant contribution to the literature. However there are several issues which need addressing. My main concerns are about some detail lacking in the methodology, and some possible errors and more detail/clarity needed for some of the statistical analysis.

Response: Thank you so much for taking your time and effort to review our manuscript. We appreciate all your valuable comments and suggestions, which are very helpful in improving the quality of our paper. All the comments have been addressed properly in the revised manuscript, which have been clearly marked in red in the paper. Please find our itemized responses below and revised manuscript in the re-submitted files.

Point 1: I wonder if you might consider changing the title to incorporate the aim around mediating variables and demographic differences (although see comment below I don’t think this actually mediation at the moment).

Response 1: Thank you for your comments. We have changed the title to “The Impact and Mechanism of Household Financial Debt on Physical Health in China”. (Line 2-3)

Point 2: Add in background and aims sentences.

Response 2: Thank you for your comments. We have added in background and aims sentences that, “In recent years, Chinese household financial debt has been growing rapidly due to the expansion of mortgage lending, and this study aims to examine the impact of Chinese household financial debt on physical health.” (Line 12-14)

Point 3: Line 19: The effect is more significant: Please consider re-wording to stroner.

Response 3: Thank you for your comments. We revised the sentence ”the effect is more stronger for those who are middle-aged, married, and have low income levels”. (Line 20)

Point 4: Please consider expanding why health is important beyond economic reasons.

Response 4: Thank you for your comments. We revised the sentence into “Health is important for human beings, and also an important means to promote sustainable socio-economic development (Sen, Anand et al. 2004).” (Line 27-28)

Point 5: Please add briefly some detail you say ‘realize a healthcare system’ in China but there is already one so please be specific here.

Response 5: Thank you for your comments. We revised the sentence into “the Chinese government attaches great importance to population health, triggering the 14th Five-Year Plan for National Health in May 2022, which plans to build a better health care system and improve population health by 2025 (Stern and Xie 2020)” (Line 34-36)

Point 6: First paragraph: Is there any evidence here you can cite on if China differs in population health to other countries?

Response 6: Thank you for your suggestion. We have added contents and cited evidence to show that China differs in population health to other countries that “China, with one-fifth of the world's population, has the largest elderly population in the world (Zhou, Wang et al. 2019). The increasing morbidity and an aging population place a considerable burden on China's health and social care systems (Gong, Wang et al. 2022).”(Line 32-33)

Point 7: Line 34: 33.19 trillion yuan: Please also report this in another currency such as US dollars for international readership. It would also be helpful to have this broken down into average per household if possible please.

Response 7: Thank you for your comments. We have revised and reported the 33.19 trillion yuan both in the currency of Chinese yuan and US dollars that “In 2020, Chinese household financial debt reaches 33.19 trillion yuan (about $4.8 trillion), 13.5 times that of 2008 (Zhang, Xu et al. 2022).” (Line 53-54) Besides, we do not found the accurate official data of average debt per household, so it is a great pity we can not report this value.

Point 8: You say that ‘Only few available literature on the relationship between debt and health”: Please consider rewording this sentence. More importantly: I don’t think this is correct. There is quite a lot of literature on debt and health, including meta-analyses in the area. Please rewrite the 3rd paragraph of the introduction overviewing the research before: At the moment this downplays the amount of research that has already been done.

Response 8: Thank you for your comments. We have rewrited this paragraph and added more literature that “Among the current research on the relationship between debt and health  (Lenton and Mosley 2008, Keese and Schmitz 2011, Sweet, Nandi et al. 2013, Dackehag, Ellegård et al. 2019) (Zurlo, Yoon et al. 2014), they have analyzed the correlation between debt and mental health, mainly using samples from developed countries. For example, Sweet et al. (2013) found that people with high financial debt were associated with more depressive symptoms in the United States (Sweet, Nandi et al. 2013). Keese (2011) used German data to find that household financial debt indicators were related to mental health and obesity (Keese and Schmitz 2011). Lenton et al. (2008) conducted a study on UK residents and found a correlation between household financial debt and mental health (Lenton and Mosley 2008). Margareta (2018) found that household financial debt and payment difficulties were detrimental to mental health in Sweden (Dackehag, Ellegård et al. 2019). Karen (2014) found that debt had a negative impact on depressive symptoms and mental health by taking the sample of middle-aged and elderly people in the United States (Zurlo, Yoon et al. 2014).” (Line 62-75)

Point 9: Line 48: “A good basis for this paper”: Please re-word.

Response 9: Thank you for your comments. We have revised the sentence into “These above previous studies from different countries have made many contributions to clarifying the impact of household financial debt on health”. (Line 76-77)

Point 10: Line 56: Please define endogeneity the first time you use it.

Response 10: Thank you for your suggestion. We have further explained endogeneity that “the existing research mainly carries out correlation analysis, but few studies consider the endogenous problems such as missing variables and reverse causality, and it is difficult to draw the conclusion of causal inference.” (Line 83-85)

Point 11: Line 64: Typo (population).

Response 11: Thank you for your suggestion. We have corrected the typo.

Point 12: Line 68 onwards: Please provide more detail about the recruitment and methods of the China Family Panel Studies.

Response 12: Thank you for your suggestion. We have revised and provided more detail about the recruitment and methods of the China Family Panel Studies that “CFPS has been conducted every two years since the official survey in 2010, and now has five years of complete survey data. For the 2010 CFPS, a multistage probability distribution was used to stratify the sample. As a result, five provinces/regions (Gansu, Guangdong, Henan, Liaoning, and Shanghai) were selected for initial over-sampling (1600 households in each, or an aggregate of 8000) to obtain regional comparisons, and another 8000 households were selected by weighting from the remaining provinces/regions to make the overall CFPS sample nationally representative (Wang, Hu et al. 2022).”(Line 164-170)

Point 13: My main issue with this section is you say about 5 time points, but it is not clear whether you are analysing all of these, and if so if the data is longitudinal i.e. if it is the same participants linked across time. Your results read like it is not and it is cross sectional, but here and throughout the paper it needs to be clearer whether your study is cross-sectional or longitudinal.

Response 13: Thank you for your suggestion. The CFPS data is longitudinal data, also known as panel data. To make it clearer, we revised in the paper that “This paper used the longitudinal data from CFPS of 2010, 2012, 2014, 2016, and 2018.” (Line 171)

Point 14: Line 78: You say about data removed for those who answered not applicable refused to answer etc were removed but it is unclear what questions/items this is referring to.

Response 14: Thank you for your suggestion. We have added more details about the removed sample that “We excluded observations that answered "unable to judge," "missing," "not applicable," "refused to answer," "don't know," or "no data" for the independent, dependent, and mediating variables used in our study”. (Line 172-175)

Point 15: Line 82: Please change ‘We use’ to ‘This survey used’.

Response 15: Thank you for your suggestion. We have made the changes. (Line 178)

Point 16: Page 83: Typo ‘physica’

Response 16: Thank you for your suggestion. We have corrected the typo.

Point 17: Line 93: Was chronic disease defined in the survey, examples given etc?

Response 17: Thank you for your suggestion. We have added more examples of chronic disease that “These chronic diseases include diabetes, hypertension, respiratory diseases, etc.” (Line 193-194)

Point 18: Line 116: Please define exogenous variable the first time you use this.

Response 18: Thank you for your suggestion. We further added to the endogeneity and the solution to the endogeneity problem that “The core variable of our study, household financial debt, may have an endogeneity problem because people with physical diseases may incur household financial debt due to excessive medical expenditures, and thus there may be an inverse causality between household financial debt and physical health. To address this endogeneity problem, we chose the instrumental variable method. The instrumental variable is the five-year annualized interest rate of China obtained through the OECD database. This instrumental variable was chosen for the following reasons: the economics theory on the equilibrium of the borrowing market suggests that the decline in the amount of household borrowing is a rational reflection of the residents’ response to lower borrowing rates (Debelle 2004), and therefore, there is a negative correlation between interest rates and household financial debt. At the same time, interest rates, also as the macro indicator, are set by the central government and can hardly be influenced by the health of a household or a resident (Frank and McGuire 2000), thus it can be considered an effective instrument variable” (Line 206-218)

Point 19: Line 120 onwards: Mental health measure of depression. Please give details here: was this a standardised measure? If not how was it developed? What was the chronbachs alpha for this sample?

Response 19: Thank you for your suggestion. Depression is a commonly used variable to represent mental health in CFPS (Jiang, Lu et al. 2020, Ao, Dong et al. 2021, Yu, Hu et al. 2022). CFPS survey on the mental health of residents, including questions such as “How often do you feel hopeless about the future” and “How often do you find it difficult to do any-thing?”. The higher the score, the higher the depression level of the residents. The sum of the scores of the 8-question depression scale in the CFPS database is used to measure mental health status. The higher the sum of the scores, the more serious the depression and the worse the mental health status of the respondents. We examined the reliability coefficient of the CES-D8 scale and the results showed that cronbach's alpha was greater than 0.7. (Line 236-237)

Point 20: Line 133: Please define hukou.

Response 20: Thank you for your suggestion. Hukou refers to household registration status. We admit that Hukou is not very easy to understand for international readers, so we changed the name of this variable to household registration. (Line 239)

Point 21: Line 138: Was there anyone not identifying as male or female, or were there only these two binary options on the survey?

Response 21: Thank you for your comment. There was not anyone not identifying as male or female. In the CFPS database, there are only two binary options for genders, male and female.

Point 22: Statistical analysis section: Please give details if the variables were normally distributed and met the criteria for normal distribution.

Response 22: Thank you for your comment. According to the law of large numbers and the law of central limit, when the sample capacity is greater than 30, the sample means will obey a normal distribution and the regression coefficients will be Best Linear Unbiased Estimators (BLUE) (Kennedy 2008).Our sample capacity is greater than 10,000, so we do not need to test whether the sample obeys the criterion of normal distribution.

Point 23: Lines 160-164: You do not need to repeat all of the means, standard deviations in the text as they are in the table. Please edit.

Response 23: Thank you for your suggestion. We have censored this paragraph, and only the means and standard deviations of the explanatory and core variables are reported. (Line 279-285)

Point 24: For categorical variables of gender (and possibly education) it does not make sense to report these as a mean when they are categorical, please report % instead, and consider if adding these as continuous variables into the regression models is the correct way to analyse these categorical variables.

Response 24: Thank you for your suggestion. We reported these variables as percentages that “49.9% of the sample is men; 72.7% of the sample is rural households; 77.4% of the sample is married.” (Line 282-285)

Point 25: For all tables: Please make clear what ‘_cons’ is.

Response 25: Thank you for your suggestion. We have changed “_cons” to “constant term”, so as to make it clear.

Point 26: For all tables: Please add in details of what control variables were added and the effect of these.

Response 26: Thank you for your suggestion. We have added in details of what control variables were added in tables. However, we did not show the regression coefficient of the control variable mainly due to the following reasons: First, the question we are studying is the impact of household debt on physical health, so we only need to pay attention to the value and significance of the coefficient of family debt. The coefficients and significance of other control variables are not important for our research conclusion. Second, there are too many tables in our paper. If the coefficients of control variables are listed, it will take up too much space, and may even exceed the maximum number of words specified by the journal. Third, to present it more clearly, we indicate the control variables in the model in the notes section of each table.

Point 27:  For all tables: I don’t think p<0.1 should be used, keep significance to p<.05 or less given the amount of tests used.

Response 27: Thank you for your suggestion. We have made the changes. We removed p<0.1 and retained only p<0.05 and p<0.01.

Point 28:  You say in methods about interest rates being included but this is not clear in the results where this fits (presumably one of the control variables but more detail is needed here).

Response 28: Thank you for your comments. We have added a section to state why instrumental variables are fitted that “In the first stage of regression, the regression coefficient of the instrumental variable interest rate is -0.026 and is significant at the 1% level, and the sign of the regression coefficient is consistent with the economic theory of equilibrium in the lending market (Debelle 2004)”. (Line 311-314)

Point 29: Table 5: This is described here and in discussion as an intermediary/mediating test. However I don’t think that it is. Here you show the link between household debt and mental health and health behaviours. This is interesting and important, but it doesn’t show, as the rest of the paper suggests, a mediating role with physical health. This could be done in another way for example through statistical mediation. Please change the results to show that health behaviours and mental health do actually have an intermediary effect on physical health, or revise the rest of the paper accordingly to show that this is a separate finding.

Response 29: Thank you for your comments. we have added a section on structural equation modeling to show the pathway diagram and tested the significance of the mediating mechanism using a mediated bootstrap test. (Line 333-359)

Point 30: Table 7 typo (psysical).

Response 30: Thank you for your suggestion. We have made the changes.

Point 31: Please revise discussion according to changes to results suggested above.

Response 31: Thank you for your suggestion. We have performed a mediating effect test and proved the significance of the mediating effect. We have revised discussion according to changes to results suggested above that “Second, household financial debt can have a negative impact on residents' health by affecting their healthcare behavior and mental health. People with debt cannot afford high medical expenses, and may choose cheap clinics with low medical level for medical treatment. Therefore, the disease may not be effectively treated, which will affect physical health. This conclusion is consistent with previous research on debt and consumer spending (Loukoianova, Wong et al. 2019). In addition, household financial debt makes residents bear the pressure of repayment and brings them mental health problems such as depression, while depression and psychological burden are often considered to be related to physical health diseases such as hypertension and obesity. This conclusion is consistent with the research on the impact of debt on mental health (Pomerleau and Pomerleau 1991) and the relationship between mental health and physical health (Ohrnberger, Fichera et al. 2017)”. (Line 416-427)

Point 32: As above, please change wording around talk of mediation.

Response 32: Thank you for your suggestion. We performed a complete mediating effect test and proved the significance of the mediating effect.

Point 33: Please be careful with language about causality e.g. line 257 (household debt has a significant negative effect on residents’. This relates to point about being clear your data is cross-sectional. Line 267-270 is another example of where the conclusions go beyond the results, as per comments above you haven’t shown that cigarettes mediate the link between mental and physical health.

Response 33: Thank you for your suggestions. Firstly, the data we used is longitudinal data. We have revised in the paper that “This paper used the longitudinal data from CFPS of 2010, 2012, 2014, 2016, and 2018,” (Line 171) Secondly, we revised the paragraph that ”People with debt cannot afford high medical expenses, and may choose cheap clinics with low medical level for medical treatment. Therefore, the disease may not be effectively treated, which will affect physical health. This conclusion is consistent with previous research on debt and consumer spending (Loukoianova, Wong et al. 2019). In addition, household financial debt makes residents bear the pressure of repayment and brings them mental health problems such as depression, while depression and psychological burden are often considered to be related to physical health diseases such as hypertension and obesity. This conclusion is consistent with the research on the impact of debt on mental health (Pomerleau and Pomerleau 1991) and the relationship between mental health and physical health (Ohrnberger, Fichera et al. 2017)”. (Line 417-427)

Point 34: Add cross-sectional to limitations.

Response 34: Thank you for your suggestion. The data we used is longitudinal data.

Point 35: Conclusions: Again be cautious about using term mediating.

Response 35: Thank you for your suggestion. We revised and performed a mediating effect test and proved the significance of the mediating effect.

Reviewer 3 Report

Let me frankly say that I found this manuscript pretty weak, esepcially in terms of analytical approach. Although the research question might be of interest (even though the main relation between household economic debt (Hd) and the physical health (PH) does not look so important in comparison to a model that takes into account many aspects of the respondents health status, simultaneously ) by concluding with the expected negative relation between HD and PH does not contribute to anything of substantially novel, even if it has been controlled, more or less, for other variables. So then, getting to the analytical approach with a basic regression model I found the authors' approach not to be very clear since they run many models without considering potential interactions. First of all with binary variables as predictors on another binary or ordinal variables you should use logistic models or logistic regression if at least one predictor is continuous and not basic regressions with metric outcomes. Secondly, you should not run separate models but one overall model with multiple predictors simultaneously and possibly checking for interactions. In light of this latter, I could not understand how the mediation role of mental health and health beahviour have been introduced in the model. There is not any path diagram of mediatated relations. Not even is clear how the instrumental variables have been introduced. The basic regression strategy proposed by the authors by running single models looked like avoiding the interaction problem among the predictors that is one of the most important issue to verify. Furthermore and given the large sample size I was wondering why the authors did not apply Structural Equation Modeling techniques instead of that poor basic regression. In this way, I do suggest to have a look at the following Bollen's paper as an example of reference: https://doi.org/10.1080/00273171.2018.1483224

Finally, there are also many concerns regarding the panel data, how did the authors control for the repeated measures across the period of time? They cannot simply pooling the data without testing for any shericity test for the homogeneity of variances and covariances across the waves of times. At least, but not last, why the authors calculated the  logarithm of net income per capita? 

Author Response

Let me frankly say that I found this manuscript pretty weak, esepcially in terms of analytical approach. Although the research question might be of interest (even though the main relation between household economic debt (Hd) and the physical health (PH) does not look so important in comparison to a model that takes into account many aspects of the respondents health status, simultaneously ) by concluding with the expected negative relation between HD and PH does not contribute to anything of substantially novel, even if it has been controlled, more or less, for other variables. So then, getting to the analytical approach with a basic regression model I found the authors' approach not to be very clear since they run many models without considering potential interactions.

Response: Thank you so much for taking your time and effort to review our manuscript. We appreciate all your valuable comments and suggestions, which are very helpful in improving the quality of our paper. All the comments raised have been addressed properly in the revised manuscript, which have been clearly marked in red in the paper. Please find our itemized responses below and revised manuscript in the re-submitted files.

Point 1: First of all with binary variables as predictors on another binary or ordinal variables you should use logistic models or logistic regression if at least one predictor is continuous and not basic regressions with metric outcomes.

Response 1: Thanks for your valuable comments.First, when ordered categorical variables are the dependent variables and the variables are grouped into categories >= 5 groups, ordered logistic regression is consistent with linear regression results (Menard 2002). Many studies similar to ours have also used linear regression models when self-rated health on a 5-point scale is the explanatory variable (Shi 2022). Secondly, to verify the stability of the results, we chose the ordered logistic regression, fixed effects-ordered logistic regression method in the robustness test section, which proved the stability of the baseline regression results of the study. (Line 299-309)

Point 2: Secondly, you should not run separate models but one overall model with multiple predictors simultaneously and possibly checking for interactions. In light of this latter, I could not understand how the mediation role of mental health and health beahviour have been introduced in the model. There is not any path diagram of mediatated relations. Not even is clear how the instrumental variables have been introduced. The basic regression strategy proposed by the authors by running single models looked like avoiding the interaction problem among the predictors that is one of the most important issue to verify.Furthermore and given the large sample size I was wondering why the authors did not apply Structural Equation Modeling techniques instead of that poor basic regression. In this way, I do suggest to have a look at the following Bollen's paper as an example of reference: https://doi.org/10.1080/00273171.2018.1483224

Response 2: Thank you for your comments. First, we added a separate section to introduce the mediating mechanism and the mediating variable. (Line 115-131) Second, we stated the reasons why the instrumental variable was introduced and why it was chosen in the research methodology section. (Line 205-218) Third, we have added a section on structural equation modeling to show the pathway diagram and tested the significance of the mediating mechanism using a mediated bootstrap test. (Line 333-359)

Point 3: Finally, there are also many concerns regarding the panel data, how did the authors control for the repeated measures across the period of time? They cannot simply pooling the data without testing for any shericity test for the homogeneity of variances and covariances across the waves of times.

Response 3: We chose the GMM (Generalized Method of Moment) method for robustness testing. The GMM method does not rely on the assumption of homoskedasticity and is able to produce the most valid estimates even in the presence of heteroskedasticity. After the GMM robust test, we proved the robustness of the baseline regression results. (Line 310-321)

Point 4: At least, but not last, why the authors calculated the logarithm of net income per capita?

Response 4: We added the reason for this in the paper that “Moreover, for income, because the range of values is too large compared to other variables, we took the logarithm of the net income per capita variable in CFPS, so as to reduce the impact on the regression results due to too large fluctuations in income.”(Line 248-251)

Reviewer 4 Report

Title: Household Debt and Physical Health: Evidence from China

This paper investigates the influence of household debt on residents’ physical health. Using a sample of China household tracking survey for the period between 2010 – 2018, authors find that there is a negative effect of household debt on the physical health status. Furthermore, the paper shows that household debt can affect residents' self-rated health through mediating variables such as mental health and health behaviors, and the effect is more significant for those who are middle-aged, married, and have low-income levels.

The paper discussing very interesting topic and provide good methods to analysis the main variables. However, the paper can benefit from number of comments to improve the quality of the paper. Here are my comments below.

-          In the introduction specifically after first paragraph, authors directly jump to talk about household debt after discussing about health and  China plan to improve health between population. However, authors should provide more information about how the health can be affected by economic situation before starting to discuss about household debts.

-          The paper focus on household debts in China, a clear motivation should be provided to show why should we focus on household debt, and why China specifically.

-            The contribution of the paper is not clear yet, for example how this paper contributes to the previous studies below:

1.       Keese M, Schmitz H. Broke, ill, and obese: The effect of household debt on health. Ruhr Economic Paper. 2011(234). 354 12.

2.       Sweet E, Nandi A, Adam EK, McDade TW. The high price of debt: Household financial debt and its impact on 355 mental and physical health. Social science & medicine. 2013;91:94-100. 356 13.

3.       Lenton P, Mosley P. Debt and health. 2008. 357 14.

4.        Dackehag M, Ellegård L-M, Gerdtham U-G, Nilsson T. Debt and mental health: New insights about the 358 relationship and the importance of the measure of mental health. European journal of public health. 2019;29(3):488-93

5.       Zurlo KA, Yoon W, Kim H. Unsecured consumer debt and mental health outcomes in middle-aged and older 374 Americans. Journals of Gerontology Series B: Psychological Sciences and Social Sciences. 2014;69(3):461-9.

-          In the contribution paragraph, the author mentioned number of points to support their contribution. However, they should emphasize and stress more on having the mediating roles of mental health and health behaviors, for people with different characteristics.

-          Authors did not provide a section for literature review and hypothesis development. This a major point to show how household debt is correlated to physical health. Furthermore, a theory to support the discussion should be provided. For example, based on what theory the authors suggest that there is a relationship between household debt and health.

-          For main explanatory variables measure, authors use a dummy variables take value of if the resident has household debt and 0 otherwise, however, this measure does not measure many perspective of debts, which could show the real association between debt and health. For example, author should examine debt structure and attitudes to debt as not all debt considers as a threat to the household.

-          Control variables, other control variables should be included such as number of children and weather the house is owned outright or owned-mortgaged.

-          Using a simple OLS could lead to biases results as the dependent variable has restricted number between 1 – 5. Authors should use Ordered Logistic Regression (OLR) approach to control for this issue.

-          It is better to show the coefficients of all control variables to provide a wide picture of the analysis. Furthermore, R-square should be presented for each model.

-          Authors should control for all type of endogeneity problems using GMM estimation method.

-          The authors mentioned that, in addition to taking the association between households and health they mentioned that they investigate the mediating role of mental health and health behaviors on the association between  households and health. However, authors did not show how they accomplished this analysis, and table 5 basically only takes the association between household debt and mental health and health behavior ( this is not mediating analysis ).  

Author Response

Response to Reviewer 4 Comments

This paper investigates the influence of household debt on residents’ physical health. Using a sample of China household tracking survey for the period between 2010 – 2018, authors find that there is a negative effect of household debt on the physical health status. Furthermore, the paper shows that household debt can affect residents' self-rated health through mediating variables such as mental health and health behaviors, and the effect is more significant for those who are middle-aged, married, and have low-income levels.

The paper discussing very interesting topic and provide good methods to analysis the main variables. However, the paper can benefit from number of comments to improve the quality of the paper. Here are my comments below.

Response: Thank you so much for taking your time and effort to review our manuscript. We appreciate all your valuable comments and suggestions, which are very helpful in improving the quality of our paper. All the comments raised by reviewers have been addressed properly in the revised manuscript, which have been clearly marked in red in the paper. Please find our itemized responses below and revised manuscript in the re-submitted files.

Point 1: In the introduction specifically after first paragraph, authors directly jump to talk about household debt after discussing about health and China plan to improve health between population. However, authors should provide more information about how the health can be affected by economic situation before starting to discuss about household debts.

Response 1: Thanks for your suggestions. We added one paragraph to provide more information about how the health can be affected by economic situation that “Health can be affected by economic situation. The relationship between health and economic status can be analyzed from the macro regional perspective and micro individual perspective (Bhargava, Jamison et al. 2001). At the macro level, the rapid economic development of a region may lead to more government financial health expenditure, and may also lead to environmental pollution and ecological damage, thus affecting the health of residents (Well 2007). Besides, at the micro level, the deterioration of personal economic status may mean the increase of work pressure, the decrease of social and economic status, and the decrease of health investment, thus affecting the health level of residents (Sweet, Nandi et al. 2013),(Xu and Wang 2022).” (Line 37-45)

Point 2: The paper focus on household debts in China, a clear motivation should be provided to show why should we focus on household debt, and why China specifically.

Response 2: Thanks for your valuable comments. We added more details about why we should focus on household debt and why China specifically that “Health can be affected by economic situation. The relationship between health and economic situation can be analyzed from the macro regional perspective and micro individual perspective (Bhargava, Jamison et al. 2001). At the macro level, the rapid economic development of a region may lead to more government financial health expenditure, and may also lead to environmental pollution and ecological damage, thus affecting the health of residents (Well 2007). Besides, at the micro level, the deterioration of personal economic stituation may mean the increase of work pressure, the decrease of social and economic status, and the decrease of health investment, thus affecting the health level of residents (Sweet, Nandi et al. 2013, Xu and Wang 2022)”. (Line 46-61)

Point 3: The contribution of the paper is not clear yet, for example how this paper contributes to the previous studies below:

  1. Keese M, Schmitz H. Broke, ill, and obese: The effect of household debt on health. Ruhr Economic Paper. 2011(234). 354 12.

  1. Sweet E, Nandi A, Adam EK, McDade TW. The high price of debt: Household financial debt and its impact on 355 mental and physical health. Social science & medicine. 2013;91:94-100. 356 13.

  1. Lenton P, Mosley P. Debt and health. 2008. 357 14.

  1. Dackehag M, Ellegård L-M, Gerdtham U-G, Nilsson T. Debt and mental health: New insights about the 358 relationship and the importance of the measure of mental health. European journal of public health. 2019;29(3):488-93

  1. Zurlo KA, Yoon W, Kim H. Unsecured consumer debt and mental health outcomes in middle-aged and older 374 Americans. Journals of Gerontology Series B: Psychological Sciences and Social Sciences. 2014;69(3):461-9.

Response 3: Thanks for your valuable comments. First, we further conducted a literature review of the contents of this literature “Among the current research on the relationship between debt and health (Lenton and Mosley 2008, Keese and Schmitz 2011, Sweet, Nandi et al. 2013, Dackehag, Ellegård et al. 2019) (Zurlo, Yoon et al. 2014), they have analyzed the correlation between debt and mental health, mainly using samples from developed countries. For example, Sweet et al. (2013) found that people with high financial debt were associated with more depressive symptoms in the United States (Sweet, Nandi et al. 2013). Keese (2011) used German data to find that household financial debt indicators were related to mental health and obesity(Keese and Schmitz 2011). Lenton et al. (2008) conducted a study on UK residents and found a correlation between household financial debt and mental health (Lenton and Mosley 2008). Margareta (2018) found that household financial debt and payment difficulties were detrimental to mental health in Sweden (Dackehag, Ellegård et al. 2019). Karen (2014) found that debt had a negative impact on depressive symptoms and mental health by taking the sample of middle-aged and elderly people in the United States (Zurlo, Yoon et al. 2014)”. (Line 62-75)

Second, we summarize the research shortcomings of this literature, which is the contribution of our study to these previous studies. “These above previous studies from different countries have made many contributions to clarifying the impact of household financial debt on health, but the existing literature may have the following problems: First, although existing studies have analyzed the impact of household financial debt on health, few have explored the intermediary mechanism and population heterogeneity. However, sorting out the intermediary mechanism of the impact of household financial debt on health and analyzing whether this correlation is different among different populations can help us understand the impact more clearly. Second, the existing research mainly carries out correlation analysis, but few studies consider the endogenous problems such as missing variables and reverse causality, and it is difficult to draw the conclusion of causal inference. Third, existing research has mainly explored the effects of family debt on mental health, while neglecting the effects on physical health. However, physical health is different from mental health (Ohrnberger, Fichera et al. 2017). Physical health is defined as the absence of disease in the organs, a high resistance to disease, and a high capacity for physical activity and work(Callahan 1973). In contrast, mental health is defined as the absence of mental illness (Rockville 1999), and living in a state of well-being (Organization 2004). Fourthly, most of the available studies have taken developed countries as their samples. However, there may be significant differences in the demand for loans, household income, consumption levels in developing countries compared to developed countries, so the findings of existing studies may not be applicable to developing countries such as China (Tsang, Von Korff et al. 2008, Balseven and Tugcu 2017)”.( Line 76-96)

Point 4: In the contribution paragraph, the author mentioned number of points to support their contribution. However, they should emphasize and stress more on having the mediating roles of mental health and health behaviors, for people with different characteristics.

Response 4: Thanks for your valuable comments. We revised and emphasized more on analyzing the mediating roles and people with different characteristics that “First, although existing studies have analyzed the impact of household financial debt on health, few have explored the intermediary mechanism and population heterogeneity. However, sorting out the intermediary mechanism of the impact of household financial debt on health and analyzing whether this correlation is different among different populations can help us understand the impact more clearly.”(Line 78-82)

Point 5: Authors did not provide a section for literature review and hypothesis development. This a major point to show how household debt is correlated to physical health. Furthermore, a theory to support the discussion should be provided. For example, based on what theory the authors suggest that there is a relationship between household debt and health.

Response 5: Thanks for your valuable comments. First, we conducted a literature review in the introduction section that “Among the current research on the relationship between debt and health (Lenton and Mosley 2008, Keese and Schmitz 2011, Sweet, Nandi et al. 2013, Zurlo, Yoon et al. 2014, Dackehag, Ellegård et al. 2019), they have analyzed the correlation between debt and mental health, mainly using samples from developed countries. For example, Sweet et al. (2013) found that people with high financial debt were associated with more depressive symptoms in the United States (Sweet, Nandi et al. 2013). Keese (2011) used German data to find that household financial debt indicators were related to mental health and obesity (Keese and Schmitz 2011). Lenton et al. (2008) conducted a study on UK residents and found a correlation between household financial debt and mental health (Lenton and Mosley 2008). Margareta (2018) found that household financial debt and payment difficulties were detrimental to mental health in Sweden (Dackehag, Ellegård et al. 2019). Karen (2014) found that debt had a negative impact on depressive symptoms and mental health by taking the sample of middle-aged and elderly people in the United States (Zurlo, Yoon et al. 2014)”. (Line 62-75)

Secondly, we added a separate section for theoretical analysis and presented the research hypotheses between Line 105-154:

“Dahlgren and Whitehead put forward the theory of social determinants of health in 1991 (Whitehead and Dahlgren 1991). This theory reckons that in addition to those factors that directly cause disease, people's social status and living and working environment determined by people's own resources are also factors affecting health, such as people's income, wealth, economic status, living conditions, etc. According to this theory, household financial debt, as a reflection indicator of people's wealth and economic status, should also be one of the social determinants of health. Therefore, this paper proposes research hypothesis 1.

H1: Household financial debt has a negative impact on physical health.

Besides, in terms of the mediating mechanism of the effect of household financial debt on physical health, this paper argues that household financial debt effect residents' physical health in 2 main ways: first, healthcare behavior. When people have household financial debt, residents will spend less on consumption (Loukoianova, Wong et al. 2019), and as a result, residents will spend less on health care. This means that residents may choose to seek medical care at cheaper, lower-quality places, and then physical health conditions may not be effectively treated, which can be detrimental to physical health. Second, mental health. Previous relevant studies have shown that family debt has a negative impact on mental health (Lenton and Mosley 2008, Keese and Schmitz 2011, Sweet, Nandi et al. 2013, Zurlo, Yoon et al. 2014, Dackehag, Ellegård et al. 2019), while additional epidemiological studies have shown that mental health indicators such as depression, stress, and poor sleep quality are significantly associated with physical health indicators such as obesity and hypertension (Ohrnberger, Fichera et al. 2017). Therefore, this paper proposes the following 2 research hypotheses.

H2: Healthcare behavior mediates the effect of household financial debt on physical health.

H3: Mental health mediates the effect of household financial debt on physical health.

Moreover, the impact of household financial debt on physical health may be het-erogeneous across populations. First, with respect to different age groups, in China, the middle-aged population is the main workforce of the family and bears the majority of the responsibility for repaying household financial debts, whereas young people and older people are in the cared-for group in China and often do not have to bear the responsibility for repaying debts (Liu, Sun et al. 2020). Therefore, the impact of household financial debt on the physical health of middle-aged people may be greater than that of young and old people. Secondly, for people with different marital status, influenced by traditional culture, Chinese people have the habit of using family debt to buy a house when they get married (Xu and Wang 2022), therefore, compared with unmarried people, married people tend to have more pressure to repay household financial debt, and the impact of household financial debt on the physical health of married people may be greater. Finally, for people at different income levels, income is an important source of funds for debt repayment (Cookson, Gilje et al. 2022), people at low-income levels may face greater pressure to repay debt, and the impact of household financial debt on physical health may be greater for people at low income levels. Therefore, this paper proposes the following three research hypotheses.

H4: Household financial debt has a greater impact on physical health in middle-aged adults compared to young and old adults.

H5: Household financial debt has a greater effect on physical health among married people compared to unmarried people.

H6: Household financial debt has a greater impact on physical health of people with lower income levels compared to those with higher income levels.”

Point 6: For main explanatory variables measure, authors use a dummy variables take value of if the resident has household debt and 0 otherwise, however, this measure does not measure many perspective of debts, which could show the real association between debt and health. For example, author should examine debt structure and attitudes to debt as not all debt considers as a threat to the household.

Response 6: Thanks for your valuable comments. Unfortunately, there are no indicators in the CFPS database that measure the structure of debt and people's attitudes toward debt. Therefore, we can't use more indicators to measure debt. We have admitted in the limitations that “more exact explanatory variables measure can be developed to reflect other perspective of debts, such as debt structure and attitudes to debt.” (Line 456-458)

Point 7: Control variables, other control variables should be included such as number of children and weather the house is owned outright or owned-mortgaged.

Response 7: Considering the availability of data, we added household size as a control variable, which is very similar to the number of children in the household. In addition, we did not add whether the house is owned outright or owned-mortgaged as a control variable. It’s mainly because in China, the way people obtain housing is mainly through home mortgage. That is, people whose house is owned-mortgaged necessarily have household financial debt. This means that people whose house is owned-mortgaged are bound to have household financial debt. Therefore, the variable whether the house is owned outright or owned-mortgaged will have a multicollinearity with the household debt variable, and we did not include it in the model.

Point 8: Using a simple OLS could lead to biases results as the dependent variable has restricted number between 1 – 5. Authors should use Ordered Logistic Regression (OLR) approach to control for this issue.

Response 8: Thanks for your valuable comments. First, when ordered categorical variables are the dependent variables and the variables are grouped into categories >= 5 groups, ordered logistic regression is consistent with linear regression results (Menard 2002),. Many studies similar to ours have also used linear regression models when self-rated health on a 5-point scale is the explanatory variable (Shi 2022). Secondly, to verify the stability of the results, we chose the ordered logistic regression, fixed effects + ordered logistic regression method in the robustness test section, which proved the stability of the baseline regression results of the study. (Line 299-309)

Point 9: It is better to show the coefficients of all control variables to provide a wide picture of the analysis. Furthermore, R-square should be presented for each model.

Response 9: Thank you for your suggestion. We added the R square of each model. In addition, we did not show the regression coefficients of control variables mainly due to the following reasons: First, the question we are studying is the impact of household debt on physical health, so we only need to pay attention to the value and significance of the coefficient of family debt. The coefficient and significance of other control variables are not important for our research conclusion. Second, there are too many tables in our paper. If the coefficients of control variables are listed, it will take up too much space, and may even exceed the maximum number of words specified by the journal. Third, to present it more clearly, we indicate the control variables in the model in the notes section of each table.

Point 10: Authors should control for all type of endogeneity problems using GMM estimation method.

Response 10: Thanks for your valuable comments. Following your kind suggestion, we chose the GMM (Generalized Method of Moment) for robustness testing. GMM does not rely on the assumption of homoskedasticity and is able to produce the most valid estimates even in the presence of heteroskedasticity. After the GMM robust test, we proved the robustness of the baseline regression results. (Line 310-321)

Point 11: The authors mentioned that, in addition to taking the association between households and health they mentioned that they investigate the mediating role of mental health and health behaviors on the association between households and health. However, authors did not show how they accomplished this analysis, and table 5 basically only takes the association between household debt and mental health and health behavior ( this is not mediating analysis ).

Response 11: Thank you for your comments. First, we have added a section on structural equation modeling to show the pathway diagram. (Line 334-359) Second, we have also added a section on testing the significance of the mediating mechanism using a mediated bootstrap method. (Line 360-369)
